# Novel RT-PCR Using Sugar Chain-Immobilized Gold-Nanoparticles Correlates Patients’ Symptoms: The Follow-Up Study of COVID-19 Hospitalized Patients

**DOI:** 10.3390/v14112577

**Published:** 2022-11-21

**Authors:** Takashi Kajiya, Hayate Sawayama, Eriko Arima, Mika Okamoto, Masanori Baba, Masaaki Toyama, Kosuke Okuya, Makoto Ozawa, Nobuhiko Atsuchi, Junichiro Nishi, Yasuo Suda

**Affiliations:** 1Clinical Research Center, Tenyoukai Central Hospital, 6-7 Izumi-cho, Kagoshima 892-0822, Japan; 2Laboratory of Collaborational Research for Glyco-Nanotechnology, Graduate School of Science and Engineering, Kagoshima University, 1-21-40 Kohrimoto, Kagoshima 890-0065, Japan; 3Center for Advanced Science Research and Promotion, Kagoshima University, Kagoshima 890-0065, Japan; 4Joint Research Center for Human Retrovirus Infection, Kagoshima University, Kagoshima 890-8544, Japan; 5Joint Faculty of Veterinary Medicine, Kagoshima University, Kagoshima 890-0065, Japan; 6Graduate School of Medical and Dental Sciences, Kagoshima University, Kagoshima 890-8544, Japan

**Keywords:** sugar chain-immobilized gold nano-particle (SGNP), follow-up study, quantitative RT-PCR (qRT-PCR), SARS-CoV-2, extraction and partial purification of RNA

## Abstract

**Background:** The transmissible capacity and toxicity of SARS-CoV-2 variants are continually changing. We report here the follow-up study of hospitalized COVID-19 patients from 2020 to 2022. It is known that the PCR diagnosis for hospitalized patients sometimes causes confusion because of the incompatibility between their diagnosis and symptoms. We applied our sugar chain-immobilized gold-nanoparticles for the extraction and partial purification of RNA from specimens for quantitative RT-PCR assay and evaluated whether the results correlate with patients’ symptoms. **Methods and Results:** Saliva specimens were taken from hospitalized patients with mild or moderate symptoms every early morning. At the time of RT-PCR diagnosis, two methods for the extraction and partial purification of RNA from the specimen were performed: a commonly used Boom (Qiagen) method and our original sugar chain-immobilized gold nanoparticle (SGNP) method. For symptoms, body temperature and oxygen saturation (SpO_2_) of patients were monitored every 4 h. **Conclusions:** It was clear that patients infected with the Delta variant needed more time to recover than those with the Omicron variant, and that the SGNP method showed more realistic correlation with the symptoms of patients compared with the common Qiagen method.

## 1. Introduction

Coronavirus disease 2019 (COVID-19), caused by severe acute respiratory syndrome coronavirus 2 (SARS-CoV-2), is still spreading worldwide [1,2,3,4]. Throughout the pandemic, the transmissible capacity and toxicity of SARS-CoV-2 variants has been changing [5]. The Delta strain spread faster than the original strain or Alpha variant, and was reported to have similar or even higher toxicity [6]. It was reported in the summer of 2021 that people in England had double the hospitalization risk with Delta than they did with the Alpha variant. The Omicron variant spread much faster than the Delta variant [7]; however, it seemed to have lower toxicity than the other variants so far [8]; the length of hospitalization, ICU admittance, and deaths were lower than during previous pandemic peaks.

We report here the follow-up study of hospitalized COVID-19 patients in Kagoshima City, Japan, from 2020 to 2022. The copy number of SARS-CoV-2 RNA in the saliva of patients was quantified by a real time reverse transcriptase–polymerase chain reaction (RT-PCR) after the partial purification of RNA from the specimen using either the Qiagen method or our original sugar chain-immobilized gold nanoparticle (SGNP) method, and compared with patients’ symptoms. Diagnosis of COVID-19 is commonly performed through the detection of SARS-CoV-2 RNA with RT-PCR testing of a nasopharyngeal swab or other specimens, including saliva. Antigen tests are generally less sensitive than PCR tests but are less expensive, and can be used at the point of care with rapid results. However, reliable biomarkers for monitoring recovery and measuring therapeutic effects have not been reported yet. For extraction and partial purification of viral RNA from specimens, the Boom method [9] has been applied as a standard method. Even though the capability of the Boom method has been improved by several companies including Qiagen, it still takes time and it also extracts and purifies free RNA of SARS-CoV-2 in specimens that have no infectious activity due to the loss of spike protein or envelope in the matured viral structure. We applied our nano-biotechnology, SGNP method [10,11,12,13,14], in which sugar chain-immobilized magnetized gold nanoparticles and micro-meter sized magnetic particles were used to collect and purify virions that have matured spike protein and viral particles. Then, by adding detergent solution to the collected mixture, viral RNAs were extracted from the purified virion, followed by applying the solution to qRT-PCR. The application of nano-biotechnology for the integration of therapy or diagnosis has been widely studied. A major application has been the usage of viral particles, which serve as excellent nano-building blocks for materials design and fabrication toward the in vitro or in vivo delivery system because of their nanometer-range size, their exceptional stability and robustness, biocompatibility, bioavailability, and so on [15]. Recently, attempts at creating nano-meter size drugs have been performed by the construction of supramolecular anti-cancer drugs [16,17]. Compared to those technologies, ours is unique and focused on the diagnostics.

According to some reports, over 80% of people with COVID-19 exhibited mild or moderate disease. It is very important to prevent severe or critical disease developing from mild or moderate disease, and also to know the timing for discharge of hospitalized patients, by careful investigation of patients with accurate diagnosis [18,19]. Monitoring viral activity in COVID-19 patients is difficult, and no established diagnostic tests are available in current clinical practice. We speculated that the SGNP method may correlate with SARS-CoV-2 viral activity and investigated to see if our SGNP method can monitor viral activity or not. In this follow-up study of hospitalized patients with COVID-19, qRT-PCR results treated with the SGNP method or the Qiagen method were compared in patients with mild or moderate disease.

## 2. Methods

### 2.1. Diagnosis of COVID-19

From April 2020 to March 2022, saliva specimens were collected from hospitalized COVID-19 patients in Tenyoukai Central Hospital while also monitoring body temperature, oxygen saturation (SpO_2_) and associated symptoms. Analysis was conducted with viral RNA extracted and partially purified from saliva specimens using real-time quantitative RT-PCR assays. Two different methods were applied for viral RNA extraction and partial purification and compared based on their correlation with symptoms. The two methods were the Qiagen method and SGNP method. The Qiagen method is a type of Boom method, in which all RNA and DNA (free RNA and DNA, and RNA from SARS-CoV-2) in the sample are extracted first and purified based on the multi-valent binding between phosphate groups of RNA/DNA and SiOH groups of silica gel on filter or beads. Qiagen kit (Cat.52904, QIAGEN GmbH, Hilden, Germany) or the compatible filter and buffer reported [9] were used. On the other hand, the SGNP method first purifies SARS-CoV-2 viral particles by capturing them with a virus-binding sugar chain-immobilized magnetized gold nanoparticle followed by collection with a second magnetic micro-particle. The sugar chain was low molecular weight dextran sulfate which have been used for porcine corona virus, influenza virus, dengue virus, and so on. The commercially available kit (SUDx-nCOV-PCR detection kit, SUDx-Biotec., Kagoshima, Japan) was used. The collected viral particles were washed with PBS buffer and destructed by the addition of detergent (0.1% SDS aqueous solution) to extract viral RNA. The extracted RNA was analyzed by quantitative RT-PCR using a commercial kit (PrimeScript RT-PCR, Takara Bio Inc., Shiga, Japan, or SUDx-nCOV-PCR detection kit, SUDx-Biotec) with Thermal Cycler Dice II or III (Takara Bio).

### 2.2. Follow-Up of COVID-19 Patients

A total of 30 hospitalized patients confirmed with SARS-CoV-2 infection showing mild or moderate symptoms in Tenyoukai Central Hospital were included. Saliva specimens were collected from patients every morning before gargling or eating breakfast. Our previous report suggested early morning saliva specimens contained higher virus load and fewer inhibitory agents than those collected at other times [14]. All samples were diluted to about 50% by the addition of PBS containing 30 mM dithiothreitol and 1% antibacterial reagents (penicillin and streptomycin) and stored in a refrigerator and/or deep freezer (−80 °C) in the buffer (PBS containing 1% PS, 10% FBS) before testing. To obtain RNA of SARS-CoV-2 for quantitative RT-PCR analyses 500 μL of the diluted saliva solution was used with the SGNP method and the final concentrated solution was 20 μL. Using the Qiagen extraction kit, 140 μL of the diluted solution was applied using the Qiagen extraction kit and the final concentrated solution was 60 μL. The extracted and partially purified RNA (1 μL) was then applied to the qRT-PCR using the same RT-PCR reagent and the same PCR apparatus. Body temperature, SpO_2_ and associated symptoms were recorded for all hospitalized patients every 4 h. This study was approved by the institutional review boards of our institutions (Kagoshima University Hospital: 160183, Tenyoukai Central Hospital: R3-8), and written informed consents were obtained from all patients.

### 2.3. Quantitative RT-PCR

For the quantification RNA of SARS-CoV-2 by RT-PCR, we selected a primer set (named mCDC set) as follows [20] according to the information of N-protein gene of SARS-CoV-2/PC00101P/human/2020/USA: Forward primer GACCCCAAAATCAGCGAAATG (21 bp, 28280-28300); Reverse primer ATGTTGAGTGAGAGCGGTG (19 bp, 28443-28425); Probe FAM-ACCCCGCATTACGTTTGGTGGACC-MGB (24 bp, 28302-28325); PCR product 164 mer. In some of experiments, we also used the following primer/probe set (named mNIID set): Forward primer ATTTTGGGGACCAGGAACTAATC (23 bp, 29120-29142); Reverse primer CGTTCCCGAAGGTGTGACTT (20bp, 29253-29234); Probe FAM-ATGTCGCGCATTGGCATGGA-MGB (20 bp, 29215-29234); PCR product 134 mer. For the RT-PCR, we used the commercially available kit (One Step PrimeScript™ RT-PCR Kit) from Takara Bio Inc. or SUDx-Biotec manufactured with Takara’s custom kit (code number, XA0236) using 1 μL of template solution in a total 16 μL reaction mixture. Thermal Cycler Dice Real Time System II or III (Takara Bio Inc.) was used as the PCR apparatus. The reaction protocol was as follows: 42 °C 300 s (reverse transcription reaction), 95 °C 10 s (hot start), 45 cycles of PCR: 95 °C 5 s, 60 °C 30 s. Copy numbers of every sample were calculated according to the calibration curve using plasmid DNA, which covers the sequence of the PCR product, and showed as copy number in 1 mL of the mixed solution of saliva and buffer. For subtyping of SARS-CoV-2 variants, we used the following primer/probe sets. Alpha variant: Forward primer: ATCAAGCCGGTAGCACAC (18 bp); Reverse primer: AAACAGTTGCTGGTGCATGT (20 bp); Probe: Cy5-CAACCCACTTATGGTGTT-BHQ3 (18 bp). Delta variant: Forward primer: CAATCTTGATTCTAAGGTTGGTGGT (25 bp); Reverse primer: GCTACCGGCTTGATAGATTTCAGTT (25 bp); Probe: Cy5-CTAAACAATCTATACCGGTAAT-BHQ3 (22 bp). Omicron variant: Forward primer: CCTTTTGAGAGAGATATTTCAACTG (25 bp); Reverse primer: GGTGACCAACACCATAAGTG (20 bp); Probe: ROX-AAACCTGCAACACCATTACA-BHQ2 (20 bp).

The reaction protocol of quantitative RT-PCR was set as follows: 42 °C 300 s (reverse transcription reaction), 95 °C 10 s (hot start), 45 cycles of PCR: 95 °C 5 s, 55 °C 30 s.

### 2.4. Culturing of Specimens

The culturing experiments were conducted in biosafety level 3 (BSL3) facilities of Kagoshima University according to the method reported with slight modification [21]. VeroE6/TMPRSS2 cell, which was highly susceptible to SARS-CoV-2 infection [22], was used for the culturing experiment of saliva specimens. The cells were cultured in Dulbecco’s modified Eagle medium (Nacalai Tesque, Kyoto, Japan) supplemented with the culture medium containing 5% heat-inactivated fetal bovine serum (FBS), 100 U/mL penicillin G, 100 μg/mL streptomycin, and 1 mg/mL G418 (Nacalai Tesque). For the positive control, SARS-CoV-2 (WK-521 strain, GISAID database ID EPI_ISL_408667), which was a clinical isolate from a COVID-19 patient and was provided by National Institute of Infectious Diseases, Tokyo, Japan [22], was used. Patient specimens mixed with buffer as described above were diluted with the culture medium and incubated with the cells for three to five days at 37 °C. After the incubation, the cytopathic effect (CPE) was monitored visually using an optical microscope or by a tetrazolium dye method.

### 2.5. Transmittance Electro-Microscopy (TEM)

Culture supernatant of MDCK cells infected with low pathogenic Avian Influenza virus (H11N6) was mixed with non-magnetized SGNP, to which the same low molecular weight dextran sulfate (DS25) was immobilized, and centrifuged at 135,000 G for 30 min at 4 °C by Optima^TM^ TLX (Beckman Coulter, Brea, CA, USA). The obtained precipitates were dissolved in PBS and mixed with ionic liquid HILEM^®^ IL1000^TM^ (Hitachi, Tokyo, Japan) according to the manual attached. Then, the mixture was placed on the TEM grid at room temperature and applied to the TEM apparatus (HT7700 TEM, Hitachi). For SARS-CoV-2, the culture supernatant of VeroE6/TMPRSS2 cells infected with SARS-CoV-2 (WK-521 strain) was incubated with 0.1% (v/v) of β-propiolactone (Tokyo Chemical Industry Co., Tokyo, Japan) overnight to inactivate viruses [23]. Then, the supernatant was mixed with the non-magnetized SGNP immobilized with DS25 and the ionic liquid, and then applied to TEM imaging similarly.

## 3. Results

Follow-Up for Patients

A total of 30 confirmed COVID-19 hospitalized patients were included. The age range was 41.8 ± 19.1 years, and 53.3% of the patients were female. At the beginning of the follow-up study, all samples except Case-2 showed positive results from both SGNP and Qiagen RNA extraction and partial purification methods.

All results of the follow-up are summarized in Table 1, where the copy number per mL of the original mixture (saliva + buffer) was calculated based on the calibration curve of RT-PCR using standard plasmid, in addition to the volume of the mixture used for SGNP or Qiagen extraction and partial purification, respectively. Day 0 in Table 1 indicates the day of hospitalization. The period between the day of onset and the day of hospitalization varied from patient to patient.

Based on the onset day for each patient, Figure 1a,b showed the diagram of SARS-CoV-2 RNA copy number of specimens for Cases 1–21 using two different partial purification methods, SGNP and Qiagen methods, respectively. Figure 1c,d are diagrams of the highest daily body temperature and the SpO_2_ (lowest oxygen saturation) of Cases 1–21, respectively.

The average of copy number of SARS-CoV-2 using SGNP and Qiagen methods were plotted with the average of the highest daily body temperature (Tmax) (Figure 1e) or with the average of the lowest oxygen saturation (SpO_2_) (Figure 1f). In Figure 1a–f, x-axis is normalized from the day of onset. It was obvious that the copy number of SARS-CoV-2 obtained by the Qiagen method were higher than those obtained by the SGNP method. The copy number using the SGNP method, and the highest daily body temperature of patients both decreased with hospitalization, suggesting recovery from COVID-19. However, the lowest SpO_2_ were almost identical, and that is probably because of mild or moderate disease patients.

In the case of Cases 9–30, we analyzed the variant viruses using our newly developed primer/probe sets for Alpha, Delta or Omicron variant, respectively. To compare Delta and Omicron variants, the daily change of the copy number of SARS-CoV-2 RNA of Case-25, -26, -27, and -29 in 1 mL of the mixture of saliva and buffer by SGNP and Qiagen methods were plotted in Figure 2a–d (x-axis is normalized from the day of onset) with the daily highest body temperature of each patient.

In all four cases, the copy number obtained by the Qiagen method were higher than those obtained by the SGNP method, displaying a similar tendency as shown in Figure 1e,f. In cases of Case-26 (Figure 2b, infected with Delta variant), the copy number obtained by the SGNP method was monitored until day 9, however, no viral RNA was found after day 8. In case of Case-29 (Figure 2d, infected with Delta variant), the copy number obtained using the SGNP method was found until day 6 and again found at day 8 and 9, however, no viral RNA was found at day 7 and after day 10. On the contrary, the copy number obtained by the Qiagen method was found until the day before discharge. Patient body temperature decreased and stayed stable after day 4. As with the present case, some studies have reported re-detectable SARS-CoV-2 tests through RT-PCR during recovery periods of COVID-19 patients, and the mechanism has not been fully clarified yet [19]. In the cases of Case-25 and Case-27 (Figure 2a,c, infected with Omicron variant), the copy number obtained using the SGNP method was found until day 3 or day 4, and that obtained by the Qiagen method was found until day 5 or 6, showing a similar tendency in that the copy number obtained by the SGNP method disappeared faster than that obtained by the Qiagen method. The body temperature of those patients stabilized after day 3.

In the case of Case-1 (Table 1), copy numbers of SARS-CoV-2 RNA extracted with the Qiagen kit did not change much from day 1 to day 10. However, the copy number determined by the SGNP method decreased day by day and was not detected after day 6. The symptoms of Case-1 disappeared from day 6, and the patient was discharged on day 11. On the contrary, in the case of Case-2, a low amount of RNA was found on day 1 by the SGNP method, and none was found by the Qiagen method. On day 3, the copy number obtained by Qiagen extraction increased. Copy numbers obtained by both methods increased from day 4, and at day 7 more copy numbers were found by both methods. In addition, the symptoms worsened and the patient was transferred to an advanced treatment hospital for COVID-19 severe disease [20].

Culturing experiments using VERO/TMPRSS2 cells were performed for selected saliva specimens obtained from COVID-19 patients in June 2021 and in September 2021 (Table 2). Thirteen selected specimens were applied for culturing experiment from Cases-6, -8, -22, -23, and -24, and their cytopathic effect was evaluated.

## 4. Discussion

It is becoming accepted that the period of illness caused by Omicron variant is shorter than that by Delta variant [7]. The follow-up study for Cases-25 to -30 reported here confirm this pattern, in which the Delta variant influenced the patients’ symptoms for longer than the Omicron variant did. However, it is also known that the transmissible capacity of Omicron variant is much higher than that of the Delta variant, which in turn is more infectious than the Alpha variant or original strain, due to the mutations in the spike protein. Most of the surfaces of cells are covered with sugar chains. It is known that most viruses first recognize sugar chains existing on the surface of a cell, bind, and then jump to the cellular high affinity receptor protein, to infect cells. The spike protein of SARS-CoV-2 possesses the binding domain for the sulfated sugar chain, heparan sulfate, on one side and the binding domain for cell surface ACE-2 receptor protein on the other side [24]. Our method for partially purifying viral RNA from a specimen containing a number of contaminants uses sugar chain-immobilized nanoparticles, where low molecular weight dextran sulfate (DS25) is covalently attached on the nano-particle. Therefore, it was thought that the nanoparticles captured the virus via the DS25. Figure 3 is a TEM image of avian influenza virus (Figure 3a) and inactivated SARS-CoV-2 (Figure 3b) captured with many non-magnetized DS-25 immobilized gold nanoparticles. It was necessary to use the non-magnetized nanoparticle, since it is impossible to see dispersed magnetized nanoparticles using a regular TEM apparatus, due to the magnetic field used in the apparatus. It is expected that the magnetized SGNP also bound to the spike protein on SARS-CoV-2 via DS-25. Since magnetized SGNP in the SUDx kit contained Fe_3_O_4_ (magnetite), the captured viruses were collected and rapidly concentrated with micro-meter sized magnetite particles in the magnetic field. At this stage of magnetic separation, viral particles were separated from the reaction solution containing the chemical reagent in the dilution buffer mixed with saliva and free RNA or DNA, and then washed with PBS. Consequently, the chemical reagent and free RNA and DNA was removed. Viral RNA was then extracted from concentrated viruses in the SGNP and micro-particle mixture, which means the SGNP method analyses the RNA from a virion which has infectivity.

The average RNA copy numbers in hospitalized patients are compared in Figure 1e. The partially purified samples by the Qiagen method had higher copy numbers than the samples treated by the SGNP method. When correlated with each patient’s highest daily body temperature, it was found that the copy number obtained by the SGNP method corresponded to patient symptoms.

The cultivation of the follow-up study involved several experimental problems; (i) specimens had to be stored in the deep freezer for certain of time since the cultivation had to be done in a BSL3 laboratory; (ii) SARS-CoV-2 lost infectious activity due to the destruction of spike protein or virus particle in the freeze–thaw process, (iii) influence of chemical reagent in the dilution buffer for the pre-treatment of saliva, (iv) factors in saliva affect the cultivation, (v) suddenly stopping the follow-up study due to the decision of the local government, and so on. At least, it was found that copy numbers obtained using the SGNP method were detected from specimens showing CPE in cultivation. In the later stages of patient hospitalization, specimens with a higher level of copy number obtained using the Qiagen method did not show CPE, while the copy number determined by the SGNP method were very low or not detected.

The SGNP method theoretically concentrates the specimen 25 times (500/20). On the other hand, the Qiagen method concentrates it 2.33 times (140/60). Therefore, in the early stages of hospitalization, the SGNP method was effective for monitoring the patient. The higher sensitivity of the SGNP method was found in the follow up study of Case-2. From the first specimen, the SGNP method was able to detect the RNA at very low levels, however, the Qiagen method did not. However, from the specimens taken during hospitalization, the RNA copy number obtained using both methods increased, and the patient symptoms were aggravated. Therefore, the patient was transferred to a specialized hospital for severe COVID-19, suggesting the importance of daily PCR testing for hospitalized patients.

From the results of the follow-up study presented here, trends of decreasing RNA concentration detected by the SGNP method were different from those detected by the Qiagen method, and correlated with patient symptoms. Therefore, the SGNP method is useful for follow-up of hospitalized patients or for preventing false-positive diagnostic results compared with the standard Qiagen method or so-called direct PCR, which cannot distinguish the RNA from the infectious virion and free RNA from destructed or immature virion. Since over 80% of COVID-19 cases exhibited mild or moderate disease in general, even in the early period of the pandemic in 2020, it is still very important to identify patients who have the potential to develop severe illness, before their physical condition deteriorates. In addition, it is also important to determine the day of discharge of patients and to suggest when patients can return to a regular lifestyle. To monitor COVID-19 infectivity and severity, quantitative RT-PCR testing using the SGNP method is useful.

### Study Limitations

This study has limitations related to study design and methods of data collection. It is a non-randomized single center study where all confounding factors and biases could not be eliminated from the picture. Sample size is small. Finally, the follow-up period of each patient is different, because the length of hospital stay is different.

## 5. Conclusions

Using RNA extraction and partial purification using SGNP, which is a magnetized gold nanoparticle where virus-binding sugar chains are immobilized to capture viral particles, followed by quantitative RT-PCR method showed more realistic correlation with symptoms of mild or moderate COVID-19 patients compared with the common Qiagen method. This novel RT-PCR using SGNP may enable monitoring viral infectivity and activity in not only COVID-19 but also other viral infection, such as influenza patients.

## Figures and Tables

**Figure 1 viruses-14-02577-f001:**
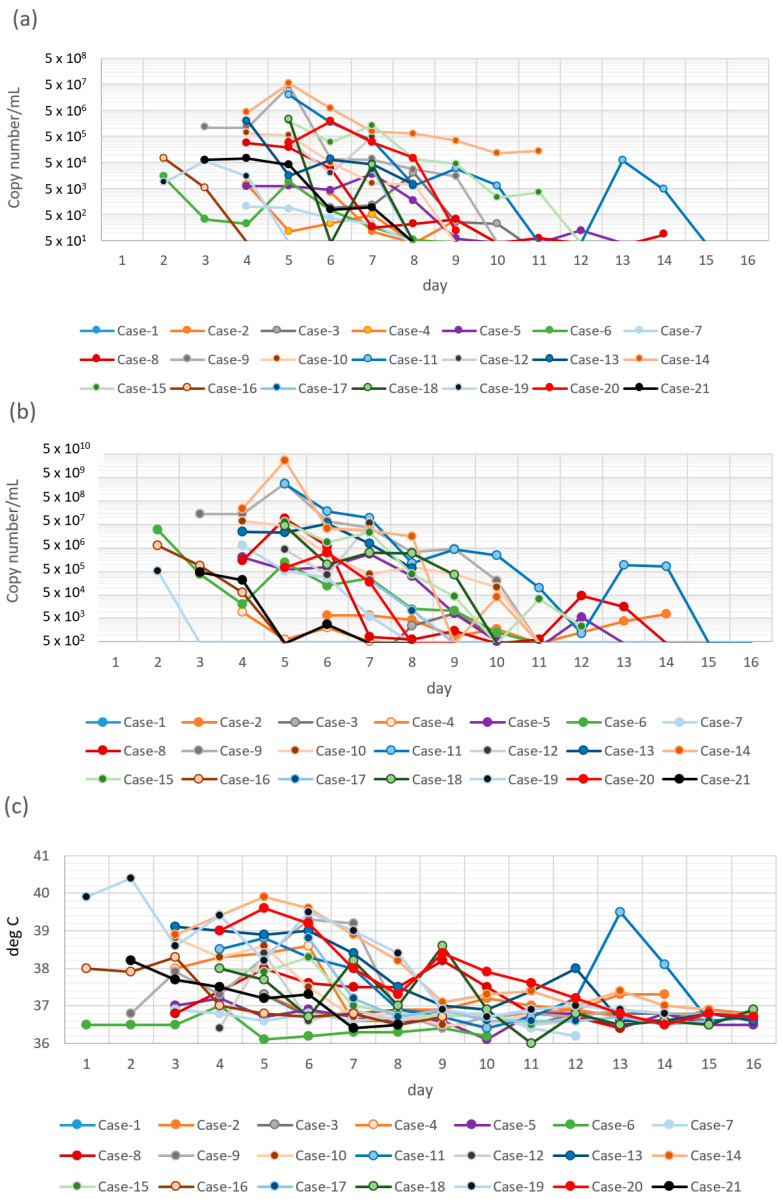
Diagram of copy number/mL of specimen mixed with buffer by SGNP method (**a**) and by Qiagen method (**b**) versus day after onset. Diagram of highest daily body temperature (**c**) and oxygen saturation (**d**) versus day after onset. The average of copy number/mL of specimen mixed with buffer by SGNP method and by Qiagen method, and average of highest daily body temperature versus day after onset (**e**). The average of copy number/mL of specimen mixed with buffer by SGNP method and by Qiagen method, and average of lowest oxygen saturation versus day after onset (**f**).

**Figure 2 viruses-14-02577-f002:**
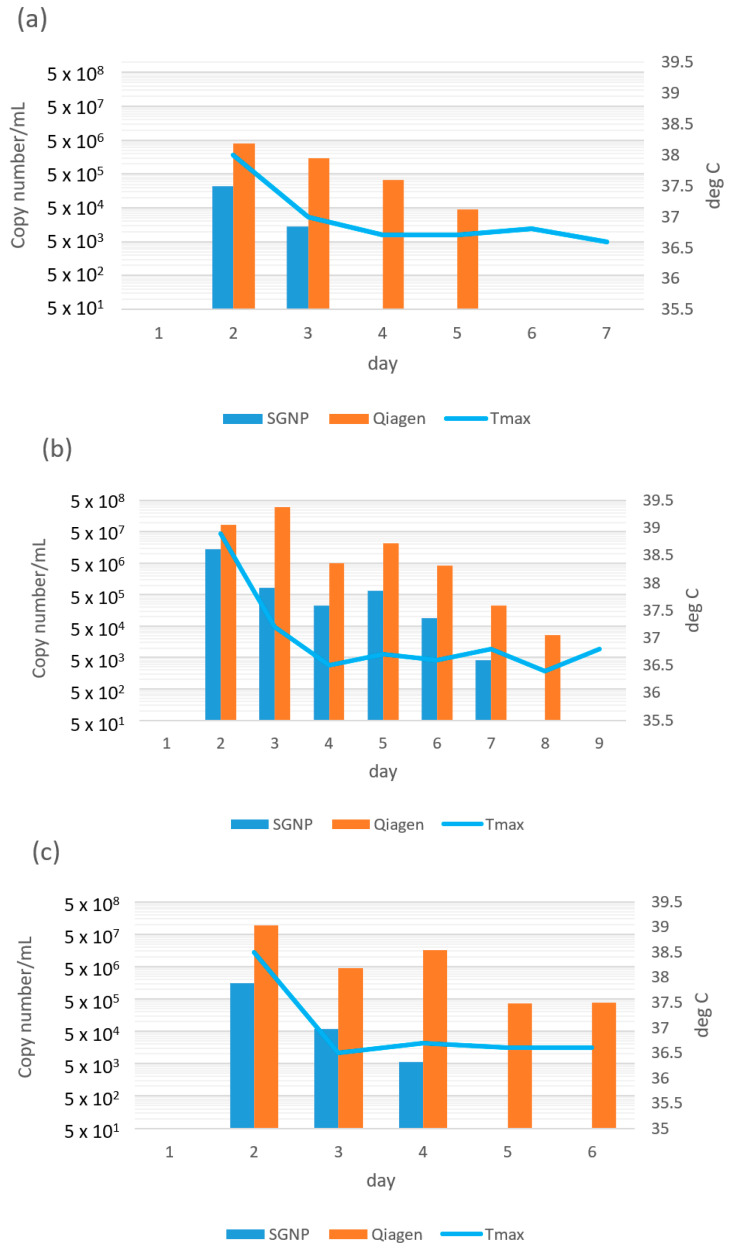
The copy number/mL of specimen mixed with buffer by SGNP method and by Qiagen method, and average of highest daily body temperature versus day after onset. (**a**) Case-25, (**b**) Case-26, (**c**) Case-27, and (**d**) Case-29.

**Figure 3 viruses-14-02577-f003:**
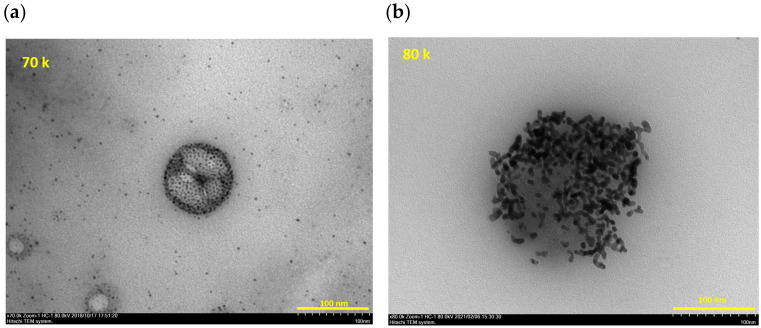
TEM image of the mixture of Avian influenza (AIV) (**a**) and inactivated SARS-CoV-2 (**b**) with DS-25 immobilized non-magnetized gold nanoparticles.

**Table 1 viruses-14-02577-t001:** Hospitalized patients and the copy number/mL from qRT-PCR, maximum daily body temperature and lowest SpO_2_ (oxygen saturation). Notes: blank column means “not done”; nd means “not detected” indicating the under the cut off value. The cut off values of copy number/mL were 50 in the case of SGNP method and 500 in the case of Qiagen method, respectively.

Case	Days Since Onset to First Sampling	Variant	Diagnostic	Day 0 (Hospitalization)	Day 1	Day 2	Day 3	Day 4	Day 5	Day 6	Day 7	Day 8	Day 9	Day 10	Day 11	Day 12	Day 13	Day 14	Day 15	Day 16	Day 17
1	5	not available	SGNP		3.4 × 10^3^	1.1 × 10^2^	4.0 × 10	3.1 × 10^2^	nd	nd	nd	nd	nd	nd							
			Qiagen		6.8 × 10^3^	6.8 × 10^3^	4.2 × 10^3^	8.6 × 10^2^	1.7 × 10^3^	nd	1.3 × 10^3^	3.8 × 10^3^	7.5 × 10^3^	nd							
			Tmax (°C)	37.3	36.8	36.8	36.9	36.9	37.2	37.0	36.9	37.3	37.3	37.3	36.8	37.0	37.1	37.0			
			SpO_2_ (min)	96	98	97	98	98	96	99	98	98	98	99	98	97	97	97			
2	4	not available	SGNP		3.4 × 10	2.9 × 10	2.8 × 10^2^	nd	nd		1.5 × 10^2^										
			Qiagen		nd	1.4 × 10^3^	4.9 × 10^2^	nd	nd		8.9 × 10^3^										
			Tmax (°C)	37.7	37.7	38.4	37.6	37.7	37.3	36.9	36.3										
			SpO_2_ (min)	96	95	96	96	96	90	99	97										
3	**5**	not available	SGNP		8.9 × 10^2^	1.1 × 10^3^	1.9 × 10^4^	2.5 × 10^2^	2.2 × 10^2^	1.7 × 10	nd		nd	nd	nd						
			Qiagen		1.8 × 10^2^	5.5 × 10	2.4 × 10^3^	7.9 × 10^3^	8.2 × 10^2^		nd		nd	nd	nd						
			Tmax (°C)	37.3	36.7	36.8	36.9	36.8	36.8	36.5	36.8	36.8	36.8	36.6							
			SpO_2_ (min)	99	97	99	99	100	100	100	99	98	98	98							
4	**3**	not available	SGNP		6.9 × 10^3^	1.1 × 10^2^	2.3 × 10^2^	4.9 × 10^2^	nd	5.0	2.0 × 10	nd	nd	nd	nd						
			Qiagen		9.4 × 10^3^	6.3 × 10^2^	1.9 × 10^3^	5.1 × 10^2^	nd	nd	1.5 × 10^2^	1.0 × 10^2^	nd	4.3 × 10	nd						
			Tmax (°C)	38.0	38.3	38.4	38.6	36.6	36.6	36.8	36.7	36.8	36.9	36.7	36.6	36.9	36.8	36.7	36.9	36.6	36.4
			SpO_2_ (min)	96	96	95	97	98	99	98	98	96	96	98	96	95	94	96	95	98	98
5	**3**	not available	SGNP		6.0 × 10^3^	6.2 × 10^3^	4.4 × 10^3^	1.8 × 10^4^	1.7 × 10^3^	5.8 × 10	nd	nd	1.3 × 10^2^	nd	nd						
			Qiagen		2.0 × 10^6^	6.0 × 10^5^	7.7 × 10^5^	2.8 × 10^6^	3.1 × 10^5^	8.0 × 10^3^	5.3 × 10^2^	2.9 × 10^2^	5.5 × 10^3^	nd	nd						
			Tmax (°C)	37.0	37.2	36.7	36.9	36.7	36.6	36.6	36.1	36.8	36.8	36.4	36.8	36.5	36.5	36.6	36.4	36.5	
			SpO_2_ (min)	97	95	98	96	97	97	98	97	98	96	97	97	98	97	96	96	98	
6	**1**	not available	SGNP		1.4 × 10^4^	3.2 × 10^2^	2.2 × 10^2^	9.1 × 10^3^	7.1 × 10^2^	1.6 × 10^2^	5.4 × 10	nd	nd	nd							
			Qiagen		3.0 × 10^7^	3.8 × 10^5^	2.0 × 10^4^	1.2 × 10^6^	1.2 × 10^5^	2.5 × 10^5^	1.2 × 10^4^	1.0 × 10^4^	1.3 × 10^3^	nd							
			Tmax (°C)	36.5	36.5	36.5	37.0	36.1	36.2	36.3	36.3	36.4	36.2								
			SpO_2_ (min)	97	96	96	97	97	98	97	96	96	97								
7	**3**	not available	SGNP		9.9 × 10^2^	8.7 × 10^2^	3.7 × 10^2^	2.2 × 10^2^	nd	nd	nd	nd	nd								
			Qiagen		6.1 × 10^6^	5.1 × 10^5^	2.6 × 10^5^	6.0 × 10^3^	nd	2.6 × 10^2^	nd	nd	nd								
			Tmax (°C)	36.9	36.8	36.6	36.8	36.6	36.7	36.8	36.8	36.4	36.2								
			SpO_2_ (min)	98	98	97	98	97	98	98	99	99	99								
8	**3**	not available	SGNP		2.7 × 10^5^	1.8 × 10^5^	2.8 × 10^4^	1.6 × 10^2^	2.2 × 10^2^	3.3 × 10^2^	nd	6.1 × 10	3.8 × 10	3.1 × 10	8.5 × 10						
			Qiagen		1.4 × 10^6^	8.8 × 10^7^	5.7 × 10^6^	8.2 × 10^2^	6.2 × 10^2^	1.4 × 10^3^	nd	6.1 × 10^2^	4.7 × 10^4^	1.5 × 10^4^	nd						
			Tmax (°C)	36.8	37.4	38.0	37.6	37.5	37.5	38.2	37.5	36.8	36.7	36.4							
			SpO_2_ (min)	97	96	96	96	96	96	97	96	96	96	97							
9	**2**	**δ**	SGNP		1.1 × 10^6^	1.1 × 10^6^	4.3 × 10^7^	6.7 × 10^4^	6.3 × 10^4^	2.6 × 10^4^	1.5 × 10^4^	nd	nd	nd	nd						
			Qiagen		1.3 × 10^8^	1.3 × 10^8^	2.7 × 10^9^	6.9 × 10^7^	3.6 × 10^7^	3.3 × 10^6^	4.5 × 10^6^	2.0 × 10^5^	nd	nd	nd						
			Tmax (°C)	36.8	37.9	37.3	38.3	39.3	39.2	36.8	36.4	36.7	36.8	36.7	36.9	36.8	36.8	36.8	36.9		
			SpO_2_ (min)	98	96	98	99	99	99	98	98	98	99	97	98	98	98	98	98		
10	3	δ	SGNP		6.6 × 10^5^	5.5 × 10^5^	5.0 × 10^4^	7.8 × 10^3^	7.4 × 10^3^	nd	nd	nd									
			Qiagen		6.6 × 10^7^	4.3 × 10^7^	2.7 × 10^6^	3.8 × 10^5^	7.5 × 10^5^	3.7 × 10^5^	1.0 × 10^5^	nd									
			Tmax (°C)	38.8	38.3	38.6	37.5	36.7	36.8	36.5	36.5	36.8									
			SpO_2_ (min)	98	97	98	98	98	98	98	98	99									
11	4	δ	SGNP		2.0 × 10^7^	1.7 × 10^6^	3.4 × 10^5^	6.6 × 10^3^	2.8 × 10^4^	6.5 × 10^3^	nd	nd	6.0 × 10^4^	4.5 × 10^3^	nd	nd					
			Qiagen		2.6 × 10^9^	1.7 × 10^8^	9.2 × 10^7^	1.1 × 10^6^	4.2 × 10^6^	2.4 × 10^6^	9.7 × 10^4^	1.1 × 10^3^	9.1 × 10^5^	8.3 × 10^5^	nd	nd					
			Tmax (°C)	38.5	38.8	38.3	38	36.9	36.7	36.4	36.7	37.2	39.5	38.1	36.6	36.7	36.5	36.8			
			SpO_2_ (min)	97	97	96	96	96	96	98	96	97	97	97	96	98	97	98			
12	4	δ	SGNP		3.2 × 10^5^	1.9 × 10^4^	4.8 × 10^5^														
			Qiagen		4.4 × 10^6^	3.5 × 10^5^	5.4 × 10^7^														
			Tmax (°C)	36.4	38.4	36.6	36.9	36.6	36.8	36.8	36.5	36.6									
			SpO_2_ (min)	99	97	98	98	98	98	98	98	100									
13	3	δ	SGNP		2.0 × 10^6^	1.5 × 10^4^	6.4 × 10^4^	4.3 × 10^4^	6.7 × 10^3^			nd	nd								
			Qiagen		2.4 × 10^7^	2.3 × 10^7^	5.3 × 10^7^	7.5 × 10^6^	6.9 × 10^5^			nd	nd								
			Tmax (°C)	39.1	39	38.9	39	38.4	37.5	37	36.9	37.4	38	36.6	36.6	36.8	36.6	35.8			
			SpO_2_ (min)	97	97	97	98	98	96	97	98	98	96	97	97	96	98	98			
14	3	δ	SGNP		4.0 × 10^6^	5.5 × 10^7^	5.9 × 10^6^	7.5 × 10^5^	6.2 × 10^5^	3.4 × 10^5^	1.1 × 10^5^	1.4 × 10^5^									
			Qiagen		2.3 × 10^8^	2.6 × 10^10^	3.3 × 10^7^	2.7 × 10^7^	1.5 × 10^7^	nd	4.0 × 10^4^	nd									
			Tmax (°C)	38.9	39.4	39.9	39.6	38.9	38.2	37.1	37.3	37.4	37	37.4	37	36.9					
			SpO_2_ (min)	98	97	98	94	95	96	97	96	98	96	97	97	98					
15	4	δ	SGNP		1.8 × 10^6^	2.9 × 10^5^	1.3 × 10^6^	6.6 × 10^4^	4.5 × 10^4^	2.3 × 10^3^	3.6 × 10^3^	nd									
			Qiagen		6.4 × 10^7^	8.5 × 10^6^	2.3 × 10^7^	3.7 × 10^5^	4.3 × 10^4^	nd	3.3 × 10^4^	2.3 × 10^3^									
			Tmax (°C)	38	37.9	38.3	37	36.8	36.7	36.8	36.5	36.7									
			SpO_2_ (min)	99	98	97	95	97	97	98	98	97									
16	1	α	SGNP		7.4 × 10^4^	5.2 × 10^3^	nd	nd	nd	nd	nd	nd									
			Qiagen		6.4 × 10^6^	8.9 × 10^5^	6.3 × 10^4^	nd	nd	nd	nd	nd									
			Tmax (°C)	38	37.9	38.3	37	36.8	36.7	36.8	36.5	36.7									
			SpO_2_ (min)	99	98	97	95	97	97	98	98	97									
17	6	α	SGNP		2.6 × 10^4^	nd	nd	nd	nd	nd	nd	nd									
			Qiagen		2.2 × 10^5^	1.1 × 10^4^	nd	nd	nd	nd	nd	nd									
			Tmax (°C)	38.8	37.2	36.7	36.9	36.6	36.6	36.6	36.7	36.5	36.6								
			SpO_2_ (min)	97	96	96	96	97	97	97	98	98	99								
18	4	δ	SGNP		2.4 × 10^6^	nd	4.0 × 10^4^	nd	nd	nd	nd										
			Qiagen		4.2 × 10^7^	9.7 × 10^5^	3.1 × 10^6^	3.0 × 10^6^	3.5 × 10^5^	nd	nd										
			Tmax (°C)	38	37.7	36.7	38.2	37	38.6	36.9	36	36.8	36.5	36.6	36.5	36.9	35.9				
			SpO_2_ (min)	98	98	97	97	97	98	97	97	98	99	98	98	98	98				
19	1	δ	SGNP		8.3 × 10^3^	5.7 × 10^4^	1.5 × 10^4^	nd	nd	nd											
			Qiagen		5.2 × 10^5^	nd	nd	nd	nd	nd											
			Tmax (°C)	39.9	40.4	38.6	39.4	38.2	39.5	39	38.4	36.9	36.7	36.9	37	36.9	36.8				
			SpO_2_ (min)	98	97	97	97	96	97	97	98	98	97	97	96	97	99				
20	4	δ	SGNP		2.7 × 10^5^	1.9 × 10^6^	2.9 × 10^5^	7.6 × 10^4^	1.2 × 10^2^												
			Qiagen		7.2 × 10^5^	3.1 × 10^6^	1.7 × 10^5^	nd	nd												
			Tmax (°C)	39	39.6	39.2	38	37.3	38.4	37.9	37.6	37.2	36.8	36.5	36.8	36.7					
			SpO_2_ (min)	97	96	96	96	98	97	97	97	96	97	97	97	97					
21	2	δ	SGNP		6.2 × 10^4^	7.0 × 10^4^	3.9 × 10^4^	7.6 × 10^2^	9.4 × 10^2^	nd											
			Qiagen		4.5 × 10^5^	2.1 × 10^5^															
			Tmax (°C)	38.2	37.7	37.5	37.2	37.3	36.4	36.5											
			SpO_2_ (min)	98	97	98	97	98	98	98											
22	4	δ	SGNP		3.7 × 10^5^	4.1 × 10^6^															
			Qiagen			3.6 × 10^7^															
			Tmax (°C)	38.2	37.5	37.4	37														
23	7	δ	SGNP		1.7 × 10^3^	1.6 × 10^3^		1.1 × 10^3^													
			Qiagen		4.6 × 10^4^	3.0 × 10^4^		6.1 × 10^3^													
			Tmax (°C)	38.5	36.8	36.8	36.2	36.6													
24	1	δ	SGNP		1.3 × 10^3^			nd													
			Qiagen		2.3 × 10^5^			nd													
			Tmax (°C)	37	37.2	36.9	36.9	37													
25	2	O	SGNP	2.2 × 10^5^	1.4 × 10^4^	nd	nd	nd	nd												
			Qiagen	4.0 × 10^6^	1.4 × 10^6^	3.4 × 10^5^	4.5 × 10^4^	nd	nd												
			Tmax (°C)	38	37	36.7	36.7	36.8	36.6												
26	2	δ	SGNP	1.4. × 10^7^	8.2 × 10^5^	2.3 × 10^5^	6.9 × 10^5^	9.1 × 10^4^	4.2 × 10^3^	nd	nd										
			Qiagen	8.3 × 10^7^	3.0 × 10^8^	5.1 × 10^6^	2.2 × 10^7^	4.2 × 10^6^	2.3 × 10^5^	2.6 × 10^4^	nd										
			Tmax (°C)	38.9	37.2	36.5	36.7	36.6	36.8	36.4	36.8										
27	2	O	SGNP		1.5 × 10^6^	6.0 × 10^4^	5.7 × 10^3^	nd	nd												
			Qiagen		9.3 × 10^7^	4.5 × 10^6^	1.6 × 10^7^	3.6 × 10^5^	3.8 × 10^5^												
			Tmax (°C)		38.5	36.5	36.7	36.6	36.6												
28	6	O	SGNP		8.1 × 10^4^	5.8 × 10^3^	nd	nd													
			Qiagen		6.9 × 10^5^	1.4 × 10^5^	2.3 × 10^3^	nd													
			Tmax (°C)	36.4	36.4	36.1	36.6														
29	1	δ	SGNP		2.3 × 10^5^	7.9 × 10^5^	1.0 × 10^5^	9.8 × 10^4^	2.8 × 10^4^	4.0 × 10^3^	nd	1.0 × 10^4^	9.4 × 10^3^	nd	nd	nd					
			Qiagen		9.4 × 10^5^	5.9 × 10^7^	8.0 × 10^6^	1.5 × 10^6^	4.3 × 10^5^	3.2 × 10^5^	9.9 × 10^4^	2.8 × 10^5^	1.8 × 10^5^	8.1 × 10^4^	6.7 × 10^4^	2.4 × 10^4^					
			Tmax (°C)		39.2	38.8	37.3	36.6	36.8	36.8	36.8	36.6	37	37.1	37	36.8					
30	5	O	SGNP	3.1 × 10^6^		1.2 × 10^4^	nd														
			Qiagen	1.2 × 10^9^		1.6 × 10^6^	nd														
			Tmax (°C)	38.9	38.1	38.1	36.9														

**Table 2 viruses-14-02577-t002:** Cultivation of selected specimens; Note: blank column means not done, other information in Table 1 is included.

Case	Days Since Onset to First Sampling	Variant	Saliva	Day 0(Hospitalization)	Day 1	Day 2	Day 3	Day 4	Day 5	Day 6	Day 7	Day 8	Day 9	Day 10	Day 11
**6**	**1**	notavailable	SGNP		1.4 × 10^4^	3.2 × 10^2^	2.2 × 10^2^	9.1 × 10^3^	7.1 × 10^2^	1.6 × 10^2^	5.4 × 10	nd	nd	nd	
Qiagen		3.0 × 10^7^	3.8 × 10^5^	2.0 × 10^4^	1.2 × 10^6^	1.2 × 10^5^	2.5 × 10^5^	1.2 × 10^4^	1.0 × 10^4^	1.3 × 10^3^	nd	
Tmax (°C)	36.5	36.5	36.5	37.0	36.1	36.2	36.3	36.3	36.4	36.2		
SpO_2_ (min)	97.0	96.0	96.0	97.0	97.0	98.0	97.0	96.0	96.0	97.0		
Culturing		CPE+			CPE−				CPE−	CPE−		
**8**	**3**	notavailable	SGNP		2.7 × 10^5^	1.8 × 10^5^	2.8 × 10^4^	1.6 × 10^2^	2.2 × 10^2^	3.3 × 10^2^	nd	6.1 × 10	3.8 × 10	3.1 × 10	8.5 × 10
Qiagen		1.4 × 10^6^	8.8 × 10^7^	5.7 × 10^6^	8.2 × 10^2^	6.2 × 10^2^	1.4 × 10^3^	nd	6.1 × 10^2^	4.7 × 10^4^	1.5 × 10^4^	nd
Tmax (°C)	36.8	37.4	38.0	37.6	37.5	37.5	38.2	37.5	36.8	36.7	36.4	
SpO_2_ (min)	97.0	96.0	96.0	96.0	96.0	96.0	97.0	96.0	96.0	96.0	97.0	
Culturing			CPE−								CPE−	CPE−
**22**	**4**	δ	SGNP		3.7 × 10^5^	4.1 × 10^6^									
Qiagen			3.6 × 10^7^									
Tmax (°C)	38.2	37.5	37.4	37								
Culturing			CPE+									
**23**	**7**	δ	SGNP		1.7 × 10^3^	1.6 × 10^3^		1.1 × 10^3^							
Qiagen		4.6 × 10^4^	3.0 × 10^4^		6.1 × 10^3^							
Tmax (°C)	38.5	36.8	36.8	36.2	36.6							
Culturing		CPE−	CPE−		CPE−							
**24**	**1**	δ	SGNP		1.3 × 10^3^			nd							
Qiagen		2.3 × 10^5^			nd							
Tmax (°C)	37	37.2	36.9	36.9	37							
Culturing		CPE+			CPE−

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
