# Peer review of "Novel RT-PCR Using Sugar Chain-Immobilized Gold-Nanoparticles Correlates Patients’ Symptoms: The Follow-Up Study of COVID-19 Hospitalized Patients"

_viruses, 2022, doi:10.3390/v14112577_

Round 1
Reviewer 1 Report
Comments to the Authors:
This manuscript applies sugar chain-immobilized gold-nanoparticles for the extraction and partial purification of RNA from specimens for quantitative RT-PCR assay, and evaluates the correlation between results and symptoms of the patients. This is an interesting and overall well-written paper. It will be a solid contribution to the viruses and will certainly appeal to many of its readers. I address some of the main issues with the manuscript in the next few paragraphs. It is recommended that this manuscript be published in viruses after completing revision.
1. The language should be improved. There are many mistakes in tables and pictures, please modify it carefully. For example, “not done?” of table 1.
2. The color of the curve in pictures are are too close to distinguish, it is recommended to draw them using the Origin software.
3. Please add the full name of the abbreviations of proper nouns in the article. When the abbreviations appear for the first time, the full name needs to be given.
4. The authors stated that “The reaction protocol of quantitative RT-PCR was modified as follows”. Please give the reasons and criteria for the selection of reaction protocol.
5. The authors stated that “In case of Case-29 (Fig. 2d, infected with Delta variant), the copy number obtained using the SGNP method was found until day 6 and again found at day 8 and 9, however, no viral RNA was found at day 7 and after day 10”. Please explain the reason for this phenomenon.
6. Please give higher-resolution TEM image to illustrate the dispersion of nanoparticles clearly.
7. How to remove the effect of chemical reagent in the dilution buffer for the pre-treatment of saliva?
8. The conclusion is not a good summary of the work and must be improved.
9. Introduction, the authors mentioned the influence and harm of virus. In order to support this statement, the following recently published important related papers should be cited: Chem. Soc. Rev. 2021, 50, 2839; Adv Mater. 2022, 34, 2106388.
Reviewer 2 Report
Authors have proposed a Novel RT-PCR method using sugar chain-immobilized gold-nanoparticles for COVID-19 patients. The article is quite interesting and designed on the emerging nanotechnology-based approach using gold-nanoparticle in follow-up patients.
In my opinion some query and suggestions are required before publishing this manuscript:
1. Authors should improve the introduction section by adding more content which supports the study design.
2. Title should be modified as it creates confusion regarding the design of the manuscript.
3. Authors must explain the statistical analysis used in this study.
4. If possible please confirm the TEM results with SEM.
5. Data incorporated in the table 1 is not visible, authors must improve the presentation of data.
6. In graph 1e, what is left Y axis represents, for ex-5.E+8……. Is this right presentation? Similarly in case of graph 1f.
7. For clarity, please improve the data of table 2.
8. Authors have selected only 30 patients in this study, please explain the limitations of this model in terms of patient size. Please explain other limitations of this model too, if any.
9. Authors are suggested to improve the conclusion section by adding future perspectives of this study.
1. The authors are advised to recheck the whole manuscript for improving the typographical and grammatical errors carefully.
Round 2
Reviewer 2 Report
Authors have improved the manuscript as per the instructions, now it can be considered for publication.